# OpenReview forum: "Soft Robot Assisted Human Normative Walking: Real Device Control via Reinforcement Learning Without a Simulator"
_ICLR.cc/2025/Conference — ICLR 2025 Conference Withdrawn Submission_

### Official Review · Reviewer_jBgV · 2024-10-27

**Soundness:** 3
**Presentation:** 3
**Contribution:** 3
**Rating:** 6
**Confidence:** 3

**Summary:**

This paper presents a system to learn controllers for robot-assisted human walking. Offline learning is used from human walking data, and online adaptation is used to further fine-tune the controller. The proposed system is demonstrated to generate controllers for a soft robotics exosuit that can help reduce human walking effort.

**Strengths:**

A system that can generate controllers that reduce human walking effort using a soft exosuit. This can impact the wider adoption of machine learning techniques on such systems.

A comprehensive analysis of the online adaption controller that shows significant improvement and personalized adaption.

**Weaknesses:**

There is no video of the results.

There is a lack of comparison to existing work. What is the state of the art in the literature regarding effort reduction with an exoskeleton, either using the same robot or other exoskeletons?

There is not much innovation in terms of methodology. Are there any limitations of the current methods? What is preventing this from scaling up as mentioned in future work? Is it just simply putting more time into doing experiments?

**Questions:**

How necessary is the initial offline phase? Can the system work by just using the online adaptation phase with some reasonable initial guess of the policy?

What is the theoretical maximum effort reduction of the assisted walk can achieve? Is the proposed method getting close to that?

---

> ### Author Response · Authors · 2024-11-25
> **We sincerely thank the reviewer for their valuable feedback. In response to the suggestion, We have now included video clips in supplementary material of all four participants walking under real-time control by AIP.**
>
> > There is no video of the results.
>
> We sincerely thank the reviewer for their valuable feedback. In response to the suggestion,We have now included video clips in supplementary material of all four participants walking under real-time control by AIP.
>
> >There is a lack of comparison to existing work. What is the state of the art in the literature regarding effort reduction with an exoskeleton, either using the same robot or other exoskeletons?
>
> The reduction of muscular effort using soft inflatable exosuits has been explored in some studies,
> though research in this area remains limited. Notably, soft inflatable knee exosuits have been employed to alleviate the muscular effort of lower limb muscles, with some studies reporting an averaged maximum reduction of 7.37\% for the VL muscle [Sridar, S et, al 2017, Sridar, S et, al 2020, Hasan, I. M et, al. 2022], compared to our method, which achieves a mean reduction of 20\%. Additionally, a soft inflatable HIP exosuit was shown to reduce muscular effort of the gluteus medius in STAND task by 30\% in [H. D. Yang et, al 2020]. But please note that this work DOESN'T present any control method to the device, and this work focuses on the STAND task.
>
> In the context of lower-limb exoskeletons, Katherine et, al 2024 use motor driven lower limb exoskeleton reduces Ankle plantarflexion effort for average 32 \% and peak vastus medialis (around knee) activity decreased with generic assistance by 13\%. Zhang, Juanjuan, et al. 2017 use motor driven ankle exoskeleton reduces soleus effort by 41 \%. Additionally other method [Steven et, al 2015, Slade, Patrick, et al. 2022] use electromyography (EMG) as a method of evaluation rather than as a primary control signal for assisted walking [Yumbla, E. Q et, al. 2021].  Metabolic costs are usually used to measure effort . The metabolic measure usually takes several minutes to collect a meaningful sample, this has made it not feasible for the problem under our consideration. Since most of exoskeletons are motor driven and  EMG/metabolic costs are different measures, it is not meaningful to compare between two studies involving exoskeleton and soft exosuit.
>
> > There is not much innovation in terms of methodology. Are there any limitations of the current methods? What is preventing this from scaling up as mentioned in future work? Is it just simply putting more time into doing experiments?
>
> Indeed, this work is not about innovating new algorithms. But the innovation of this work is a creative solution with demonstrated success in several important aspects. For instance, how to achieve offline policy when there is no simulator is an open question since most of current offline-to-online frameworks address sim-to-real issue. One of the key contributions of this work is achieving a walkable offline policy without relying on a simulator or dynamic model, and once the offline policy achieved, theoretically all offline-to-online framework can do its job.
>
> Another critical issue is human trust. Without trust, developing a successful model becomes significantly more challenging [Brenneis et,al. 2021]. In this application, the absence of a simulator made it difficult to gain participant trust and address concerns about the method. For this initial step in applying RL without a simulator, we chose to leverage a familiar offline algorithm (BC) and online algorithm (dHDP). This approach helped build participant confidence, which is essential for experiments requiring human-robot interaction.
>
> Scaling up this approach is not simply a matter of putting more time into experiments. Similar to medical testing, where initial studies focus on proving the effectiveness and safety of a treatment before expanding to larger trials, our priority is to first establish that the proposed method works reliably under normaltive walking. This involves demonstrating that the approach can achieve meaningful results, such as reducing EMG effort and algorithm convergence. Once this foundational proof of concept is established, scaling up would involve addressing additional challenges, such as variability across different users, devices, and tasks. This step would require further validation through broader studies, potentially involving more diverse participant groups, longer-term trials, and different robotic systems.

---

> > ### Author Response · Authors · 2024-11-25
> > **cont.**
> >
> > >How necessary is the initial offline phase? Can the system work by just using the online adaptation phase with some reasonable initial guess of the policy?
> >
> > Having an initial guess of the policy is certainly beneficial and can be used for initializing online adaptation. However, deploying this initial guess into the system presents significant challenges. Hardcoding the policy directly into neural network weights is impractical due to the complexity and flexibility required for neural network representations. Even with an initial guess of the policy, it is necessary to train the actor's weights to accurately represent this policy, which requires an offline training phase. The offline training phase serves as a critical step to encode the initial policy into the neural network while ensuring it is properly parameterized and ready for further online adaptation.
> >
> > Directly learning an optimal control using dHDP is potentially possible as discussed earlier that dHDP is an online learning method and has been demonstrated to successfully learn multiple challenging tasks directly online. However for the current problem, minimizing experiment time involving human users motivates us to develop a cost-efficient offline policy first, followed by dHDP online learning.
> >
> > >What is the theoretical maximum effort reduction of the assisted walk can achieve? Is the proposed method getting close to that?
> >
> > We thank reviewer for asking this interesting question. Directly calculating a theoretical reduction of EMG is really challenging. To the best of our knowledge, no literature exists for computing this value
> > because the muscles always have a baseline non-zero EMG activity that varies from person to person. However we could at least calculate the expected torque contribution that the exosuit provides by comparing the biological torque with the assistive torque from the exosuit as a rough estimate of how much lesser force the user exerts, since muscle force is directly related to EMG activity. For different weight, for example, we have seen different "theoretical" reduction as:\\
> > 1. For an 80 kg subject: 19\% of the biological extension can be supported/provided by the exosuit.\\
> > 2. For a 50 kg subject: 30\% of the biological extension can be supported/provided by the exosuit.
> >
> >
> > EMG activity is directly related to the amount of muscular force exerted, so this provides a rough insight into how much the muscle will benefit, although quantitatively, the \% reduction in torque does not directly translate to \% reduction in EMG. This is because the torque received by human knee is less than the output of the soft actuator. For example when the leg is straight, even though the soft actuator is fully inflated, the torque received by knee is much less than the actuator can generate. As we stated in the above som studies reporting an averaged maximum reduction of 7.37\% for the VL muscle [Sridar, S et, al 2017, Sridar, S et, al 2020, Hasan, I. M et, al. 2022], compared to our method, which achieves a mean reduction of 20\%.

---

> > > ### Author Response · Authors · 2024-11-25
> > > **Reference on Our Comment**
> > >
> > > Sridar, S., Nguyen, P. H., Zhu, M., Lam, Q. P., \&amp; Polygerinos, P. (2017). Development
> > > of a soft-inflatable exosuit for knee rehabilitation. 2017 IEEE/RSJ International
> > > Conference on Intelligent Robots and Systems (IROS), 2017-Septe, 3722–3727.
> > > https://doi.org/10.1109/IROS.2017.8206220
> > >
> > >  Sridar, S., Poddar, S., Tong, Y., Polygerinos, P., \&amp; Zhang, W. (2020). Towards
> > > Untethered Soft Pneumatic Exosuits Using Low-Volume Inflatable Actuator Composites
> > > and a Portable Pneumatic Source. IEEE Robotics and Automation Letters, 5(3),
> > > 4062–4069. https://doi.org/10.1109/LRA.2020.2986744
> > >
> > >  Hasan, I. M., Yumbla, E. Q., \&amp; Zhang, W. (2022). Development of a Soft Inflatable
> > > Exosuit for Knee Flexion Assistance. 2022 9th IEEE RAS/EMBS International
> > > Conference for Biomedical Robotics and Biomechatronics (BioRob), 1–6.
> > > https://doi.org/10.1109/BioRob52689.2022.9925474
> > >
> > >  H. D. Yang, M. Cooper, T. Akbas, L. Schumm, D. Orzel and C. J. Walsh, \&quot;A Soft
> > > Inflatable Wearable Robot for Hip Abductor Assistance: Design and Preliminary
> > > Assessment,\&quot; 2020 8th IEEE RAS/EMBS International Conference for Biomedical
> > > Robotics and Biomechatronics (BioRob), New York, NY, USA, 2020, pp. 692-699, doi:
> > > 10.1109/BioRob49111.2020.9224283
> > >
> > >  Yumbla, E. Q., Qiao, Z., Tao, W., \&amp; Zhang, W. (2021). Human Assistance and
> > > Augmentation with Wearable Soft Robotics: a Literature Review and Perspectives.
> > > Current Robotics Reports, 2(4), 399–413. https://doi.org/10.1007/s43154-021-00067-0
> > >
> > >  Dylan JA Brenneis, Adam S Parker, Michael Bradley Johanson, Andrew Butcher, Elnaz
> > > Davoodi, Leslie Acker, Matthew M Botvinick, Joseph Modayil, Adam White, and Patrick M Pi-
> > > larski. Assessing human interaction in virtual reality with continually learning prediction agents
> > > based on reinforcement learning algorithms: A pilot study. arXiv preprint arXiv:2112.07774,
> > > 2021.
> > >
> > > Poggensee, Katherine L., and Steven H. Collins. "Lower limb biomechanics of fully trained exoskeleton users reveal complex mechanisms behind the reductions in energy cost with human-in-the-loop optimization." Frontiers in Robotics and AI 11 (2024): 1283080.
> > >
> > > Zhang, Juanjuan, et al. "Human-in-the-loop optimization of exoskeleton assistance during walking." Science 356.6344 (2017): 1280-1284.
> > >
> > > Collins, Steven H., M. Bruce Wiggin, and Gregory S. Sawicki. "Reducing the energy cost of human walking using an unpowered exoskeleton." Nature 522.7555 (2015): 212-215.
> > >
> > > Slade, Patrick, et al. "Personalizing exoskeleton assistance while walking in the real world." Nature 610.7931 (2022): 277-282.

---

### Official Review · Reviewer_tHbP · 2024-11-04

**Soundness:** 3
**Presentation:** 3
**Contribution:** 3
**Rating:** 8
**Confidence:** 3

**Summary:**

This paper tackles the key problem of how to make the robot work effectively with humans without having a computer simulation to test things first - a significant challenge since most robotics work relies heavily on simulation before real-world testing.  The paper presents a systematic investigation of the problem and shows that their method can successfully train soft robots to aid walking without relying on simulation, opening new possibilities for human-robot interaction in assistive devices.

The authors propose a two-step approach they call AIP (Adaptation from Imitating expert Policy). First, they learn from normal human walking patterns without the robot's assistance. Then, they use this as a starting point for real-time learning where the robot adapts to each individual person's walking style. Their method focuses on improving data quality through a technique called RIIV (Reducing Intra-person and Inter-person Variation) and ensures safety through careful constraint design. The researchers validate their approach with both theoretical analysis and practical experiments on four subjects, demonstrating reduction in muscle effort while maintaining natural walking patterns.

----
*Post Rebuttal*: Updated the Score.

**Strengths:**

- The paper presents a clear, well-structured methodology for direct learning on physical devices, with precise problem formulation and experimental validation.

- The data-driven approach (AIP) is simple yet effective, demonstrating successful learning without simulation through a combination of offline imitation and online adaptation.

- Strong empirical results with clear metrics (20% EMG reduction) and thorough ablation studies validating design choices.

**Weaknesses:**

- The investigation lacks analysis of system degradation effects (e.g., soft actuator wear and tear) on the learned timing shifts discussed in Q3. This raises questions about long-term reliability.

- The paper would benefit from a baseline comparison showing human adaptation alone without robot learning, to isolate the benefits of the co-adaptation process.

- Limited discussion on the generalizability of the approach - specifically, which components are specific to soft exosuits (like RIIV data processing) versus which are generally applicable to human-robot learning systems.

- While the paper demonstrates successful learning without simulation, it doesn't fully address whether this approach eliminates the need for simulation in this domain. Have we achieved oracle performance? A discussion of the domain specific need of simulation and tradeoff to direct learning approaches would strengthen the contribution.

**Questions:**

NA

---

> ### Author Response · Authors · 2024-11-25
> **We sincerely thank the reviewer for their valuable feedback. In response to the suggestion, we have included videos as supplementary material through the provided attachment.**
>
> > The investigation lacks analysis of system degradation effects (e.g., soft actuator wear and tear) on the learned timing shifts discussed in Q3. This raises questions about long-term reliability.
>
> We appreciate the reviewer's concern regarding the analysis of system degradation effects, such as soft actuator wear and tear, on the learned timing shifts. However, investigating degradation effects aligns more closely with manufacturing research / industral scale evaluation and falls outside the primary focus of our study, which is centered on developing reinforcement learning methods for controlling soft wearable robotics.
>
> Moreover, the airbag components in the exosuit are replaceable, allowing for straightforward maintenance after a period of usage. This replacement strategy helps mitigate the impact of degradation, ensuring consistent system performance over time.
>
> In terms of timing shifts due to wear and tear, all 5 participants used the same setup and thus all have experienced natural wear and tear. As such, learning has adapted to wear and tear if there is an effect which causes time shift.
>
> > The paper would benefit from a baseline comparison showing human adaptation alone without robot learning, to isolate the benefits of the co-adaptation process.
>
> Thank you for this very interesting question. We have included a new Figure 8 in Appendix H, where we kept a learned  policy unchanged and conducted an evaluation session to analyze human adaptation independently. From Figure 8, we observe that human adaptation from trial to trial is quite limited, with no clear trend in the timing variable. However, we observe significant variation across participants. For instance, Participant 1 shows a delayed timing on $t_A$, whereas other participants do not exhibit this adaptation. Similarly, Participant 4 demonstrates a slight delay in timing on $t_C$, which is not observed in others.
>
> This observation aligns with intuition. For example, when learning to ski, individuals adapt differently—some may quickly stand on the skis and take off, while others struggle to find balance or even to stand up. Similarly, in our formulation, without a robot to guide the human user, humans have difficulty reacting meaningfully to the robot. However, with policy updates that provide feedback via human interaction with the exosuit, humans can perceive the difference and gain a sense of direction for adapting their responses more effectively as we showed on Figures 4 and 7.
>
> > Limited discussion on the generalizability of the approach - specifically, which components are specific to soft exosuits (like RIIV data processing) versus which are generally applicable to human-robot learning systems.
>
> Thank you to the reviewer for raising this question, which helps us further clarify and improve the potential of our approach. In our current method, the formulation of state, action, and cost is specific to a class of soft exosuit device and the task goal. For other devices and tasks, these formulations will need to be adapted based on the physical insights on the problem and task specifications.
>
> For RIIV, the intra-person processing step (Equation 4) aims to capture invariant characteristics of the task, while the inter-person processing step (Equation 5) is based on well-established biomechanics literature [Zhang et, al. 2020b] to define a reasonable range for human knee joint movement. In other human-robot interactive systems, the principles of RIIV should still apply to identifying invariant task-related characteristics.
>
> In terms of extracting the offline policy from a human demonstration, we utilized a motion capture system, which is widely employed in human-robot walking related studies. Our approach does not need to involve many participants, neither requires a prolonged data collection process.
>
> The approach of using motion capture data as ground truth of typical human walking control policy, while depends on sensor data as policy inputs, can be readily adapted to other applications. Additionally, the offline-to-online framework itself is well-established in the robotics learning field, although most implementations are within a sim-to-real construct. Our work extends this paradigm to a real-world setting without reliance on a simulator, demonstrating its broader applicability.

---

> > ### Author Response · Authors · 2024-11-25
> > **cont.**
> >
> > >While the paper demonstrates successful learning without simulation, it doesn't fully address whether this approach eliminates the need for simulation in this domain. Have we achieved oracle performance? A discussion of the domain specific need of simulation and tradeoff to direct learning approaches would strengthen the contribution.
> >
> > Thank you for this important perspective.
> > We do not intend to eliminate the need for simulation in this domain. Simulators and dynamic models provide significant advantages for learning in robotics applications. For instance, simulations can avoid distracting  factors such as sensor and actuator noise, communication delays, and actuator delays, etc. As a result, this simplifies policy convergence. Using simulations also eliminates logistical challenges such as arranging participant visits, saving considerable time. Furthermore, simulations can generate abundant state-action tuples/trajectories per second, whereas real-world data collection is much more expensive, slower, and costly. In our study, it typically yields only about 100+ data tuples for approximately 10 minutes of walking. On the other hand, the sim-to-real approach may take longer learning time overall than online programming including the time for developing the simulation and testing on the robot. As is well understood, it is a real challenge to accurately represent the real world, which naturally includes uncertainties such as variations in the environment [Inoue, Tadanobu, et al. 2017].
> >
> > As noted in the Related Work and Introduction sections (lines 49–62 and 135–137), prior to this work, no simulator-free learning methods existed for achieving stable and robust control of the exosuit. This gap arises from the fundamental challenge of modeling interaction dynamics between the user and the robot for optimal coordination. This paper partially answers a critical real-world question: how to deploy RL in applications where no simulator is available. We demonstrate this showing how we should work with the physics of the problem first in order to get meaningful results.
> >
> > We cannot guarantee oracle performance due to several complex factors. First, the definition of optimal performance varies depending on the chosen performance metrics. In our approach, we focused on measurable metrics such as kinematic errors (e.g., peak knee angle error) and EMG effort. However, if additional factors, such as human confidence or trust of the device, were incorporated into the formulation, the optimal performance might shift accordingly. Second, from a learning perspective, longer training periods typically result in more optimal solutions. However, when working with real humans, there are inherent physical limitations, making extended training sessions not feasible. Training for hours is challenging and raises practical and ethical concerns. What we can confidently state is that, under the current solution framework, our approach effectively reduces EMG effort for all participants. As recognized by Reviewer tHbP " Strong empirical results with clear metrics (20\% EMG reduction) and thorough ablation studies validating design choices.”

---

> > > ### Author Response · Authors · 2024-11-25
> > > **Reference on Our Comment**
> > >
> > > Li Zhang, Geng Liu, Bing Han, Zhe Wang, Yuzhou Yan, Jianbing Ma, and Pingping Wei. Knee joint
> > > biomechanics in physiological conditions and how pathologies can affect it: A systematic review.
> > > Applied Bionics and Biomechanics, 2020:1–22, 4 2020b. ISSN 1176-2322. doi: 10.1155/2020/
> > > 7451683.
> > >
> > >  Inoue, Tadanobu, et al. "Deep reinforcement learning for high precision assembly tasks." 2017 IEEE/RSJ International Conference on Intelligent Robots and Systems (IROS). IEEE, 2017.

---

> ### Comment · Reviewer_tHbP · 2024-11-26
> **Post Rebuttal**
>
> The Authors adequately address most of the concerns raised in the review.
>
> - Addition of frozen policy to investigate human learning was very welcome.
>
> - (may be for future work) I would be nice to hav some Experimental results demonstrating the system's claimed ability to handle degradation effects in real-time. This would provide a compelling advantage over purely offline-trained systems, which typically struggle to adapt to unexpected system changes. Such validation would significantly strengthen the contribution of the work.
>
> - Simulator Integration Discussion: I understand that the current work is not trying to argue against the importance of simulators, rather focusing on the question of what it takes to deploy a simulator free learning system. Having said that a section with some discussions pertaining to simulator could be interesting addition.
>    - What would be the minimum fidelity requirements where having a simulator to provide meaningful benefits for offline training. Are there any metrics that the simulator will have to be able to "match" before being useful for offline training ?
>    - What are the current state-of-the-art simulation platforms that may be able to simulate these systems.
>    - Technical challenges specific to simulating this particular setup.
>
> Including these elements may enhance the paper's practical impact and provide valuable insights for future research directions in this area.

---

> > ### Author Response · Authors · 2024-11-27
> >
> > > The Authors adequately address most of the concerns raised in the review.
> >
> > Thank you. We are pleased that the reviewer is satisfied with most of our responses. Below, we provide point-by-point replies to your thoughtful and inspiring questions. It has been a pleasure engaging in such discussions with you.
> >
> > > Addition of frozen policy to investigate human learning was very welcome.
> >
> > Thank you for bringing up this important point in the first place. We're glad to provide such data.
> >
> > > (may be for future work) I would be nice to have some Experimental results demonstrating the system's claimed ability to handle degradation effects in real-time. This would provide a compelling advantage over purely offline-trained systems, which typically struggle to adapt to unexpected system changes. Such validation would significantly strengthen the contribution of the work.
> >
> > We appreciate your further elaborating on the point. Now we can see your point better. Indeed, this is a great point of consideration for a future study. What we can think of is that we "artificially" create some degradations in the soft exosuit, varying in degrees, and then again starting from the offline policy as the initial policy for online learning. So then we will be investigating two important aspects of our solution approach. 1) Can dHDP adapt to accommodate the degradations, and what is the breaking point if the degradation is too severe? To address this, experiments may take place at discrete degradation levels, and be given sufficient time for dHDP to adapt. 2) For a "typical" and "gradual" degradation, how fast can dHDP adapt, and more importantly, keep up with the degradation. 3) Can offline policy capture some of the degradation effect and continue adapting online?
> >
> > To conduct a new experiment like this we will need some substantial preparation, from a new protocol to perform systematic design/evaluation/testing to quantify the degradations, and to final experimental testing of offline and online investigations, among other tasks. It is going to take some great amount of effort to implement and carry out the study. But the reviewer has brought up a very important aspect for RL control to deal with real life challenges naturally present in the physical environment. We like your idea very much. Thank you again for bringing it up!
> >
> > >simulator Integration Discussion: I understand that the current work is not trying to argue against the importance of simulators, rather focusing on the question of what it takes to deploy a simulator free learning system. Having said that a section with some discussions pertaining to simulator could be interesting addition.
> >
> > Yes, indeed, this is a great suggestion and we'll include a discussion pertaining to simulators and their role in real life RL control applications. We believe including this could help provide a even better context of our current work, and also may shed some light for future studies involving simulators.  We again appreciate your constructive feedback.

---

> > > ### Author Response · Authors · 2024-11-27
> > > **cont.**
> > >
> > > >What would be the minimum fidelity requirements where having a simulator to provide meaningful benefits for offline training. Are there any metrics that the simulator will have to be able to "match" before being useful for offline training ?
> > >
> > > Great question. Our thoughts are along the following lines.
> > >
> > > The two aspects, "the minimum fidelity requirements" and "metrics", for the simulator to match go hand-in-hand. Both should be based on  the physics of the problem and the needs of the application. In order to determine "the minimum fidelity requirements" and "metrics", we need to understand the physics of the problem in order to model at a level that is commensurate with the application's needs. For example, the flight control and the cart-pole control should be treated differently. On the one hand, the physics of the two problems differ, on the other hand, the fidelity requirements for flight control simulation should be much more stringent than those for cart-pole control simulation for obvious reasons. Accordingly, the metrics for the two problems can be quite different again albeit that the specific measurements are physical and thus the respective fidelity metrics are different.
> > >
> > > To be more specific,
> > > consider the Apache helicopter as an example. FLYRT was developed by Boeing with multiple years of effort. It is a sophisticated and realistic flight simulation model of the Apache helicopter. As flight control testing is expensive, pouring resources to develop high fidelity model is natural. The simulation model in this case can significantly reduce real life test needs. In FLYRT case, if offline training can be shown successful in FLYRT while taking into account different flight conditions including difficult conditions and environmental uncertainties, then we can expect the sim-to-real transfer will only require some controller fine tuning albeit the many in between steps with increasing testing conditions to approach real life.
> > >
> > > But not all applications can afford developing near real life, high fidelity models as Boeing did to develop FLYRT. Some general guidelines may be useful in gauging what appropriate levels of sophistication some models can be expected.
> > >
> > > For the simulation, some fidelity requirement are:
> > >
> > > 1. Physical fidelity: The simulator must accurately replicate the dynamics and kinematics of the real system, including mass properties, friction coefficients, and actuator models. Inaccurate modeling can lead to a reality gap where policies trained in simulation fail in the real world [Jakobi, Nick et, al 1995, Oberkampf, William L. et, al 2004, Sargent, Robert G 2010].
> > >
> > > 2. Equipment fidelity:  The simulator can emulate or replicate the equipment being used, which includes all hardware components of the system [Liu, Dahai, et al. 2008].
> > >
> > > 3. Motion fidelity: It is defined as the degree to which a simulator can reproduce the sense of motion in the operational environment [William L. et, al 2004, Liu, Dahai, et al. 2008].
> > >
> > > Before utilizing a simulator for offline training, certain metrics should be evaluated to ensure its usefulness.
> > >
> > > The following metrics may help evaluate the usefulness of a model.
> > >
> > > 1. System identification errors: one common metric to evaluate simulation model and real world data is using system identification to quantify the difference between simulated and real-world system responses. Lower errors suggest better model accuracy [Ljung, Lennart 2010].
> > >
> > > 2. Sensor data errors: compare the data between simulation and real-world data. May use the mean average error (MAE), mean square error (MSE) or mathematical statistics to evaluate the difference [Jakobi, Nick et, al 1995].
> > >
> > > 3. Sensitivity analysis:  systematic investigation of the reaction of model outputs to drastic changes in model inputs and model structure [Kleijnen, Jack PC 2010].

---

> > > > ### Author Response · Authors · 2024-11-27
> > > > **cont**
> > > >
> > > > >What are the current state-of-the-art simulation platforms that may be able to simulate these systems.
> > > >
> > > > The majority of the current SOTA simulation platforms, such as Gazebo, MuJoCo, PyBullet, NVIDIA Isaac Sim, Unity and Unreal Engine, are designed for industrial robots or  human-robot interaction w/o robot physically attached to human (refer to our global response).
> > > >
> > > > Even though they are quite well-sophisticated and frequently used in research, but challenges persist in sim-to-real transfer.  Potentially directed efforts may help address some of the following issues pertaining to their respective applications.
> > > >
> > > > 1) The Reality gap: it includes differences in dynamics, sensor noise and environmental interactions [Salvato, Erica, et al. 2021].
> > > >
> > > > 2) Complex hardware mechanisms: the actuator dynamics, the delays in control signals introduced by multiple hardware and software layers, the low-level controller dynamics, and compliance/ damping at the joints. According to Hwangbo, Jemin, et al. 2019 "these mechanisms are nearly impossible to model accurately".
> > > >
> > > > 3) Limitations in sensor simulation: Accurately modeling sensor characteristics, including noise profiles and distortions, is challenging. Vision sensors, in particular, are affected by lighting conditions, reflections, and occlusions that are hard to replicate in simulation.
> > > >
> > > > 4) Computational constraints: High-fidelity simulations are computationally intensive, limiting the speed of training and the ability to simulate large-scale environments or long-duration tasks. Although currently NVIDIA Isaac can provide GPU support for this issue, considering  the GPU price, the learning cost for high fidelity simulation are really high.
> > > >
> > > > 5) Safety and robustness: Ensuring that policies are safe and robust when faced with unexpected real-world events is a significant concern, especially for autonomous systems operating in dynamic environments [Garcıa, Javier et, al 2015].
> > > >
> > > >
> > > > For wearable robotics such as robotic exoskeleton and prothesis, the SOTA simulation platform is OpenSim (which we discuss in more details below.)
> > > >
> > > > > Technical challenges specific to simulating this particular setup.
> > > >
> > > > For simulation platforms, such as gazebo, MuJoCo, PyBullet, NVIDIA Isaac Sim, Unity and Unreal Engine, they are primarily designed for robotics and physics-based simulations, with a focus on modeling RIGID BODY dynamics, articulated mechanisms, and environmental interactions. Simulating muscle and neural reflexes involves modeling soft tissue dynamics, muscle contractions, and neural feedback loops, which are more specialized and fall outside the core functionalities of those general-purpose physics engines.
> > > >
> > > > For simulating human locomotion, perhaps the most relevant platform is OpenSim [Delp, Scott L., et al. 2007], which is a specialized open-source software platform explicitly designed for modeling and simulating the biomechanics of human and animal movement. It excels in capturing the complex dynamics of musculoskeletal systems, including muscle forces, joint mechanics, and neural control.
> > > > However, it lacks in some important aspects. For example, the contact force simulation between two objects (such as the ground reaction force between the foot and the ground surface), can be unrealistic at times. It is estimated based on Hunt-Crossley model which is unable to prevent objects from clipping through each other, for example, the foot goes into the ground by penatrating the ground surface. More generally,  OpenSim is not designed for soft body dynamics or large deformations, which are central to soft actuator (including soft exosuit) behavior. Modeling the non-linear material properties of soft actuators (e.g., silicone, elastomers) requires advanced finite element methods, which are not natively supported in OpenSim.
> > > >
> > > > Another key challenge for simulating our system and other human wearable robots is the dynamic interaction between the human wearer and the robot creates a highly coupled and complex system, making it even more challenging to model accurately (please refer to lines 812 to line 819). On top of that, human physiological and physical conditions change even within a day or time of the day. Such changes can be very different for different individuals as we all differ from one another. Overcoming all these challenges may not be a matter of time and resource, but some fundamental scientific discoveries are needed.

---

> > > > > ### Author Response · Authors · 2024-11-27
> > > > > **reference for the followup comment**
> > > > >
> > > > > Jakobi, Nick, Phil Husbands, and Inman Harvey. "Noise and the reality gap: The use of simulation in evolutionary robotics." Advances in Artificial Life: Third European Conference on Artificial Life Granada, Spain, June 4–6, 1995 Proceedings 3. Springer Berlin Heidelberg, 1995.
> > > > >
> > > > > Oberkampf, William L., Timothy G. Trucano, and Charles Hirsch. "Verification, validation, and predictive capability in computational engineering and physics." Appl. Mech. Rev. 57.5 (2004): 345-384.
> > > > >
> > > > > Sargent, Robert G. "Verification and validation of simulation models." Proceedings of the 2010 winter simulation conference. IEEE, 2010.
> > > > >
> > > > > Liu, Dahai, et al. "Simulation fidelity." Human factors in simulation and training. CRC Press, 2008. 91-108.
> > > > >
> > > > > Ljung, Lennart. "Perspectives on system identification." Annual Reviews in Control 34.1 (2010): 1-12.
> > > > >
> > > > > Kleijnen, Jack PC. "Sensitivity analysis of simulation models: an overview." Procedia-Social and Behavioral Sciences 2.6 (2010): 7585-7586.
> > > > >
> > > > > Tobin, Josh, et al. "Domain randomization for transferring deep neural networks from simulation to the real world." 2017 IEEE/RSJ international conference on intelligent robots and systems (IROS). IEEE, 2017.
> > > > >
> > > > > Tremblay, Jonathan, et al. "Training deep networks with synthetic data: Bridging the reality gap by domain randomization." Proceedings of the IEEE conference on computer vision and pattern recognition workshops. 2018.
> > > > >
> > > > > Jiang, Yifeng, et al. "Simgan: Hybrid simulator identification for domain adaptation via adversarial reinforcement learning." 2021 IEEE International Conference on Robotics and Automation (ICRA). IEEE, 2021.
> > > > >
> > > > > Zhang, Jingwei, et al. "Vr-goggles for robots: Real-to-sim domain adaptation for visual control." IEEE Robotics and Automation Letters 4.2 (2019): 1148-1155.
> > > > >
> > > > > Salvato, Erica, et al. "Crossing the reality gap: A survey on sim-to-real transferability of robot controllers in reinforcement learning." IEEE Access 9 (2021): 153171-153187.
> > > > >
> > > > > Hwangbo, Jemin, et al. "Learning agile and dynamic motor skills for legged robots." Science Robotics 4.26 (2019): eaau5872.
> > > > >
> > > > > Garcıa, Javier, and Fernando Fernández. "A comprehensive survey on safe reinforcement learning." Journal of Machine Learning Research 16.1 (2015): 1437-1480.
> > > > >
> > > > > Delp, Scott L., et al. "OpenSim: open-source software to create and analyze dynamic simulations of movement." IEEE transactions on biomedical engineering 54.11 (2007): 1940-1950.

---

### Official Review · Reviewer_1QdU · 2024-11-04

**Soundness:** 2
**Presentation:** 3
**Contribution:** 2
**Rating:** 5
**Confidence:** 4

**Summary:**

This paper introduces a method for controlling a soft robotic exosuit to assist human walking.  The key idea is to use reinforcement learning to directly learn a control policy from real-world interactions between the human and the exosuit.  This is achieved through an offline-to-online approach called Adaptation from an offline Imitating expert Policy. In the offline phase, the robot learns to imitate human walking patterns from expert demonstrations.  This provides an initial policy for the online phase, where the robot personalizes the assistance by fine-tuning the controller through direct interaction with the user.  The method is evaluated with four participants, showing that it can effectively reduce muscle effort during walking while maintaining a normal gait.

**Strengths:**

1. The paper addresses the challenging problem of controlling soft exosuits for human assistance, which involves complex dynamics and human-robot interaction.
2. The method is designed to work directly in the real physical environment, without relying on a simulator, which is often difficult to construct accurately for soft robots and human-robot interaction. This makes the approach more practical and potentially more effective than sim-to-real methods.

**Weaknesses:**

1. The method used in this paper is naive BC and policy gradient, however, there are other related offline to online frameworks such as [1], and the authors are encouraged to perform experiments with other methods.
2. The experiments are conducted with a small number of participants. It is unclear how well the method would generalize to a larger and more diverse population, including people with different walking patterns, or physical impairments.
3. Missing of hardware details.
4. Missing comparison with previous methods in exosuits, does reinforcement learning really help?
5. This article is more suitable for the robot conference or journal since it focuses on field robots rather than general methods.

[1] Uni-O4: Unifying Online and Offline Deep Reinforcement Learning with Multi-Step On-Policy Optimization, Kun LEI and Zhengmao He and Chenhao Lu and Kaizhe Hu and Yang Gao and Huazhe Xu, ICLR 2024

**Questions:**

See weakness.

---

> ### Author Response · Authors · 2024-11-25
> **We thank the reviewer for all of their time and insighful feedback. Please see our Global Response above, which gives a comprehensive summary of this work and review discussion.**
>
> > The method used in this paper is naive BC and policy gradient, however, there are other related offline to online frameworks such as [1] ([Kun LEI, et, al. 2024]), and the authors are encouraged to perform experiments with other methods.
>
> The reviewer's point is well taken. However, we would like to stress that the AIP method succeeded within the scope of this study by using the well established BC and dHDP methods, and more importantly, WITHOUT extensive training on a simulator or utilizing a dynamic model of the human-exosuit system to obtain an offline policy. To the best of our knowledge, no existing offline-to-online frameworks, including [Kun LEI, et, al. 2024], have demonstrated directly learning from physical environment using LIMITED data for an offline policy. Additionally, as we discussed in the Introduction, it is not feasible and unrealistic to obtain a dynamic model or a simulator in our problem setting. Yet in [Kun LEI, et, al. 2024], it requires learning from a simulator, thus the method in [Kun LEI, et, al. 2024] and AIP are not comparable. A comparison between the two is not meaningful. More generally speaking, sim-to-real methods are not feasible in the current problem setting.
>
> Finally we'd like to quickly re-iterate the point that gaining participants' trust for them to interact with a (foreign) robotic device is essential in carrying on the experiments. Being able to walk while wearing a device is not as easy as one may imagine. Using simple yet effective methods with good track records such as BC and dHDP has helped participants gain trust in controlling the devices.
>
> In the revised manuscript, we will expand our discussion on existing offline-to-online methods (including [Kun LEI, et, al. 2024]).
>
> > The experiments are conducted with a small number of participants. It is unclear how well the method would generalize to a larger and more diverse population, including people with different walking patterns, or physical impairments.
>
> We have thoroughly address this issue in "Global Response". We kindly refer the reviewer to the point "On Diversity of Participants".
>
> > Missing of hardware details.
>
> We have included these details in Appendix D, we also provided detailed design in lines 174–175. And we have highlighted all the important hardware components by bold facing them for easy identification.
>
> >Missing comparison with previous methods in exosuits, does reinforcement learning really help?
>
> As we have discussed in the Introduction, there is no currently existing automatic control method available for knee assistance in soft exosuits similar to the one used in this study. The only existing method, as described in [5], requires EACH participant to walk normally on a treadmill while an expert collects data and MANUALLY designs the control system. This approach is not only time-consuming but also achieves limited results, reporting an
> averaged maximum EMG reduction of 7.37\% for the VL muscle for a well tuned controller. Our AIP method, on the other hand, achieves a mean reduction of 20\% with significantly less time and not relying on human expertise to tune the controller. In summary, yes, RL really helps.
>
>
> For other control methods applied to different types of soft exosuits such as hip suits and ankle suits, we have included them in our related work (lines 135 to 149). These methods are predominantly motor-driven cable system (in contrast, ours are based on air bags). Those cable driven systems heavily rely on specific hardware and robot design. This strong dependence makes it particularly challenging to implement them in our system and perform a fair comparison. In essence, a comparison is not meaningful. Note that, these motor driven cable systems [Li, Qinjian, et al 2022,  Li, Zhijun, et al. 2022, C. Siviy et al. 2020] can be modeled based on motor torque dynamics which make model based control method applicable. However as we discussed in the related work (lines 812 to 819), such model-based approaches are entirely not feasible for soft exosuit as generating such model is impractical or nearly impossible.
>
> This is another motivation why we want to use RL control. In contrast to the model based method, our model-free reinforcement learning approach overcomes these limitations. As long as human data can be collected for offline learning, our method can extract an offline policy to initialize online learning. This reinforcement learning approach has the potential for much broader applications in the soft exosuit field.

---

> > ### Author Response · Authors · 2024-11-25
> > **cont.**
> >
> > > This article is more suitable for the robot conference or journal since it focuses on field robots rather than general methods.
> >
> > We are unsure about how the reviewer came to this conclusion, and what the bases are for this conclusion. On the contrary,
> > according to the ICLR Call for Papers (https://iclr.cc/Conferences/2025/CallForPapers), the conference considers a broad range of subject areas, including applications in robotics. This paper showcases RL for control in scenarios where no simulator or dynamic model is available. Using a soft robotic suit, we demonstrate how to conceive an offline policy and use it to generate a walkable policy for online RL training. Just as acknowledged by the same reviewer, this is a key strength of our work: 'The method is designed to work directly in the real physical environment, without relying on a simulator, which is often difficult to construct accurately for soft robots and human-robot interaction. This makes the approach more practical and potentially more effective than sim-to-real methods.'

---

> > > ### Author Response · Authors · 2024-11-25
> > > **Reference for Our Comment**
> > >
> > > Uni-O4: Unifying Online and Offline Deep Reinforcement Learning with Multi-Step On-Policy Optimization, Kun LEI and Zhengmao He and Chenhao Lu and Kaizhe Hu and Yang Gao and Huazhe Xu, ICLR 2024
> > >
> > >  Li, Qinjian, et al. "Fuzzy-Based Optimization and Control of a Soft Exosuit for Compliant Robot–Human–Environment Interaction." IEEE Transactions on Fuzzy Systems 31.1 (2022): 241-253.
> > >
> > >  Li, Zhijun, et al. "Human-in-the-loop control of soft exosuits using impedance learning on different terrains." IEEE Transactions on Robotics 38.5 (2022): 2979-2993.
> > >
> > > C. Siviy et al., "Offline Assistance Optimization of a Soft Exosuit for Augmenting Ankle Power of Stroke Survivors During Walking," in IEEE Robotics and Automation Letters, vol. 5, no. 2, pp. 828-835, April 2020, doi: 10.1109/LRA.2020.2965072.
> > >
> > >  Sridar, Saivimal, et al. "Towards untethered soft pneumatic exosuits using low-volume inflatable actuator composites and a portable pneumatic source." IEEE Robotics and Automation Letters 5.3 (2020): 4062-4069.

---

> ### Comment · Reviewer_1QdU · 2024-11-25
> **Response to Authors**
>
> Thanks for the authors' response to my questions. I agree with the authors that sim-to-real methods are not feasible in the current problem setting, however, the paper I mentioned does not require learning from a simulator, it proposed a general framework of offline-to-online training. Therefore, I still encourage the authors to do some comparison. I take the point that the experiments in this paper have certain diversity and thank the authors for providing the hardware setting and experiment videos. I thank the authors offering more discussion about exosuits. I raised my score to 5.

---

> ### Author Response · Authors · 2024-11-25
> **Response to the "not require learning from a simulator"**
>
> We thank the reviewer again for reading our paper and for providing important feedback.
>
> We are happy that we have addressed several important concerns the reviewer had, such as
> 1) the "sim-to-real methods are not feasible in the current problem setting
> 2) ...the experiments in this paper have certain diversity.
>
> We are also glad our explanation and highlighted information are well received by the reviewer, such as 1) hardware setting, 2) experiment videos, and 3) more discussion about exosuits.
>
> Regarding the paper by Kun LEI, et, al. 2024, and its reliance on a simulator, we meant that the reported results in the paper (Figure 1,3,4) and benchmark testing (Table 1,2) were based on extensive simulations.
> Please refer to Figures 1 and 4 in the paper. Their algorithms converged in about  $100 * 5e3$ to $200 * 5e3$ environment steps.
> That is not realistic in our problem setting.
> As we have reported in our paper, first we don't have a simulator, second we only have limited data to work with because we can only ask human participants to walk certain number of steps, which are typically about  150 gait samples. We cannot have an experiment protocol approved even if we ask participants to walk 7500 steps, not half million or 1 million steps.
>
> To have a quantitative sense of the two paradigms, please refer to Figure 2 in our paper, our method converged at about Episode 50, which approximately corresponds to 7500 ($150 * 50$) environment steps in [Kun LEI, et, al. 2024]. Yet clearly, they are far away from converging. Having limited data and without extensive simulation are exactly our focus, and achieving successful learning under such stringent conditions is one of our major contributions.
>
>
> This is exactly why we stated in our Related Work section, lines 122 to 124, that "growing evidence has shown the potential of substantial performance improvement in imitation learning by merely modifying the data collection process [Belkhale et al. (2024)]". We developed our RIIV-based, data centric approach not only reduces intra-person step length variations but also reduces inter-person variation. Because of our innovative data-centric approach, plus our use of well-established methods (BC and dHDP), our AIP approach demonstrated its impressive performance as we have reported.
>
> Please also note that  [Kun LEI, et, al. 2024] uses a PPO variant for both offline to online learning. PPO and SAC are widely used in robotic learning based on extensive simulations or under structured environment. Both algorithms have shown great ability to learn stochastic policies by parameterizing a probability distribution over actions. These distribution-based policies facilitate exploration by sampling actions during training. One of the key contributions in [Kun LEI, et, al. 2024] is that "Ensemble BC approach learns diverse behavior policies that are more likely to cover all modes present in the dataset." However, in wearable robotics, particularly with humans in the loop, safety and comfort are of paramount importance. Human participants become frustrated if the robot operates erratically (due to stochastic policy) and it may not be a good idea to  diversify behavior policies.
> An analogy is how people feel when they learn to ski. Falling to the ground in arbitrary ways make them feel frustrated, a mental state that adversely affect their natural behavior or even cause injuries.
> A stochastic and more diverse behavior policy can become a source of causing discomfort to human users.
>
> In essence, our problem is a real time control problem, which is different from the extensive simulation based control problems as in the paper brought up by the reviewer. It should not be too difficult to understand this from another real time control application such as flight control. We cannot afford erratic control actions due to safety concerns. Perhaps because of this, to the best of our knowledge, none of existing soft exosuit use PPO and SAC type algorithms.
>
> Finally, We want to emphasize again that we do not claim any contribution on the GENERAL offline to online framework and as we discussed in the global response we are really happy to explore other offline to online frameworks. However since this is the VERY FIRST study of its kind in RL and in soft exosuit, we want to validate the method, show its reliability with a well-designed and executed study, and prove its value via concrete evidence before we systematically fine tuning the methods to make the offline-to-online learning even more efficient and effective.
>
> Again, we thank the reviewer for reading our paper and for the stimulating questions. We hope the reviewer become excited about our work just as we are about this innovative study after reading our further explanations on what we did and how we did it.

---

### Official Review · Reviewer_YG6k · 2024-11-10

**Soundness:** 2
**Presentation:** 3
**Contribution:** 3
**Rating:** 6
**Confidence:** 4

**Summary:**

The authors present online Adaptation from an offline Imitating expert Policy (AIP) algorithm to learn a personalized and generalizable exoskeleton controller for human subjects. The knee-assist exoskeleton-suit helps reduce the effort required by the human user. The key to their approach is to first learn an offline behavior cloned policy which acts as a good starting point for online adaptation. To boost learning performance, the paper also introduces RIIV which reduces intra and inter-person variations thus improving data quality.

**Strengths:**

The paper is written with sufficient details. The results presented are interesting and does justify the approach as a practical solution to robot assisted locomotion. Successfully getting online reinforcement learning to work on safety-critical human-robot interaction task is commendable.

**Weaknesses:**

Including a supplementary video would be extremely useful for the reader to evaluate the qualitative performance of the policy. For example, noticing how gait looked like with the exoskeleton active. (I understand the privacy concerns with human subjects involved, perhaps blurring out the users could be an option?)
Adding a section on the limitations of the proposed approach would be valuable.

**Questions:**

1) If I understand correctly, the policy only controls the onset timing and duration and not the actual air pressure of the robot. Does the air pressure used in the exoskeleton vary with different users?
2) Related to the above question, the authors mention that data from only one person was used for collecting data for the offline imitation learning phase of the algorithm. Which among the 4 users contributed to this data? or was there a different person? if so, what were the anthropometric data look like?
3) In section 3.5 - the following statement " Unlike traditional RL formulations that aim to
maximize the expected reward, in wearable robotics, the objective is to minimize the overall cost
over policy π" is not very clear, how are these two different. One can also reformulate a cost function to be a reward function. On a related note, the algorithm used for online adaptation direct heuristic dynamic
programming (dHDP) is not one of the more popular online RL algorithms, was any other approaches like PPO, SAC etc.. considered? If so, what are the pros and cons?

---

> ### Author Response · Authors · 2024-11-25
> **We thank the reviewer for all of their time and insighful feedback. Please see our Global Response above, which gives a comprehensive summary of this work and review discussion.**
>
> > Including a supplementary video would be extremely useful for the reader to evaluate the qualitative performance of the policy.
>
> We sincerely thank the reviewer for their valuable feedback.
> We have now included video clips in supplementary material of all four participants walking under real-time control by AIP.
>
> >Adding a section on the limitations of the proposed approach would be valuable.
>
> Point well taken, thank you. We had a brief discussion at the end of the Conclusion section alluding to follow-up studies based on the current results.  Now with this reviewer's suggestion and also other reviewer's questions, there is a need to clarify and now we've done that to reflect on the following points. 1) On scaling up to task complexity, even though we have demonstrated the feasibility of AIP  to directly learn in physical environment of exosuit robot control, and even though both BC [Michael Bain et, al. 1995] and dHDP [si et, al 2001] have the potential to scale up, the performance of AIP in assistive locomotion tasks such as walking at varying speed, on different surfaces, ascending and descending stairs are yet to be systematically studied. 2) On extending AIP enabled assistive walking from unimpaired to the impaired population, it is expected that a new control performance measure, a new design of offline data collection, and a new experimental protocol, among others, are required to develop a feasible control solution.
>
> >  If I understand correctly, the policy only controls the onset timing and duration and not the actual air pressure of the robot. Does the air pressure used in the exoskeleton vary with different users?
>
> 1) On control policy, yes, it only controls the onset timing and duration of the soft exosuit and for all participants. Unlike rigid exoskeletons where the torque is generated by electrical motors and can be directly used as a control parameter, for soft inflatable actuators, the amount of assistive torque is determined by both human knee angle and the actuator pressure, the two collaborative sources. It is therefore not feasible to use torque directly as the control variable for the exosuit. Instead, only properly timed inflation and deflation of the exosuit will provide the necessary and optimal assistance to the human user. Please refer to Section 3.2 for more information.
>
> 2) We keep the pressure setpoint the same for all participants. Specifically, the soft exosuit inflates from 0 up to 206.8 kpa (a safety limit as discussed in line 228). This pressure setpoint level corresponds to a maximum of 25\% knee torque for an average person (note that the actually received torque varies according to the knee-angle.) As such, we can focus on timing and duration which are more critical and dynamic to the performance of the exosuit.
>
> > Related to the above question, the authors mention that data from only one person was used for collecting data for the offline imitation learning phase of the algorithm. Which among the 4 users contributed to this data? or was there a different person? if so, what were the anthropometric data look like?
>
> Correct, the offline data was from one person (Participant 1). Please refer to Appendix E, lines 928-929 or the caption of Figure 3 that “Participant 1 provided the offline policy”, refer to line 442.
>
> The anthropometric data of all participants are summarized below.
>
> | Subject | Gender  | Age (year) | Weight (kg) | Height (cm) |
> |---------|---------|------------|-------------|-------------|
> | 1       | M       | 26         | 76          | 175         |
> | 2       | F       | 27         | 52          | 154         |
> | 3       | M       | 28         | 79          | 165         |
> | 4       | F       | 31         | 57.5        | 158         |
> | 5       | M       | 28         | 80          | 172         |
>
> > In section 3.5 - the following statement " Unlike traditional RL formulations that aim to maximize the expected reward, in wearable robotics, the objective is to minimize the overall cost over policy $\pi$" is not very clear, how are these two different. One can also reformulate a cost function to be a reward function.
>
> Point well taken.
> We meant that cost and reward are mathematically interchangeable, as cost can be represented as negative reward.
> In the context of the current problem a "cost" representation of control objective is more appropriate as physically it makes sense to reduce kinematic error and the reduced physical effort also leads to a "cost" reduction.
>
> We will rephrase the sentence in the revised paper to be more clear - "Based on the nature of the physical problem, we have formulated the control objective as to minimize the cost which is made up of the kinematic error and the physical effort. "

---

> > ### Comment · Reviewer_YG6k · 2024-11-26
> >
> > Appreciate the rebuttal. I think the changes made improve the paper and I will increase my rating to a 6.

---

> > > ### Author Response · Authors · 2024-11-27
> > >
> > > Thank you. We are pleased that the reviewer is satisfied with our responses. Your comments have helped us greatly to make improvements. We truly appreciate your feedback.

---

> ### Author Response · Authors · 2024-11-25
> **Cont.**
>
> >  On a related note, the algorithm used for online adaptation direct heuristic dynamic programming (dHDP) is not one of the more popular online RL algorithms, was any other approaches like PPO, SAC etc.. considered? If so, what are the pros and cons?
>
> Indeed, the dHDP, which was developed over two decades ago [si et, al 2001], is a less popular method than PPO and SAC. It is based on the idea that the actor adjusts the policy in the direction of the action-value gradient, and the critic updates the action-value function. This idea is considered a predecessor in a better known algorithm, the NFQCA [Hafner et,al. 2011], which in turn, is considered a predecessor of the even better known DPG algorithm  [Silver et, al. 2014]. In a nutshell, the dHDP is a bare-bone policy gradient method. All these methods can be made more stable by integrating experience replay and target networks, which were first introduced in the deep Q-networks (DQN) [Mnih et, al. 2013]. A more detailed discussion on dHDP and its performance evaluation in comparison to DDPG, as well as its several significant applications in complex and realistic engineering systems, such as
> Apache helicopter stabilization, tracking, and reconfiguration control [Enns, et, al. 2003], power grid [Sun, Jian, et al. 2012], chemical processes [Yang, Qinmin, et al. 2021], fuzzy control system [Ying, et al 2016], Single-Axis Servo Mechanism System [El-Sousy, Fayez FM, et al. 2024], and more can be found in [Wu et al. 2024].
>
> PPO and SAC are widely used in robotic learning based on extensive simulations or under structured environment. Both algorithms have shown great ability to learn stochastic policies by parameterizing a probability distribution over actions. These distribution-based policies facilitate exploration by sampling actions during training. However, applying PPO and SAC in exosuit robot that does not have a simulator or a dynamic model of human-robot interaction, presents true challenges.
>
> In wearable robotics, particularly with humans in the loop, safety and comfort are of paramount importance. Human participants become frustrated if the robot operates erratically. An analogy is how people feel when they learn to ski. Falling to the ground in arbitrary ways make them feel frustrated, a mental state that adversely affect their natural behavior or even cause injuries. A stochastic policy can become a source of causing discomfort to human users. In other real time control applications such as flight control, we cannot afford erratic control actions either due to safety concerns. Perhaps because of this,  to the best of our knowledge, all existing applications of PPO and SAC rely on simulators, which enable unlimited data collection, and controls are conducted in structured environments. The reliance on huge amount of data and the needs of safety assurance have significantly narrowed the pool of potentially feasible methods.
>
> However dHDP has been shown convergent theoretically and  demonstrated convergence under limited data conditions  empirically in several wearable robotics applications [Wen et, al. 2017a, Wen et, al. 2017b, Wen et, al. 2017c, Wen et, al. 2019, Wu et,al. 2021]. Additionally, dHDP has been shown its compatible performance to DDPG in many DMC robotic control problems [Wu et,al. 2024]. These nice theoretical properties and practical successful studies have given us the confidence to deploy dHDP in our current study, which directly learn the exosuit control solution with 5-dim state space and 4-dim action space.
>
> In summary, given the track record of dHDP in human-robot locomotion control applications, and also to build initial trust in human participants for this new soft robot control problem, we took the first step of using dHDP to learn in real time for soft robot assisted human walking with reduced effort. This has helped us overcome the critical issue to not rely on a mathematical model or simulation model, which are nearly impossible to obtain [Brenneis, et. al 2021].
> While this approach has met our expectation in the current study, we recognize the limitations or potential challenges as summarized in the Limitations section, and we plans to systematically explore and evaluate different solution architectures and algorithms in future studies.

---

> > ### Author Response · Authors · 2024-11-25
> > **Reference for our comment**
> >
> > Babič, Jan, et al. "Challenges and solutions for application and wider adoption of wearable robots." Wearable Technologies 2 (2021): e14.
> >
> >  Yue Wen, Andrea Brandt, Ming Liu, He Huang, and Jennie Si. Comparing parallel and sequential
> > control parameter tuning for a powered knee prosthesis. In 2017 IEEE International Conference
> > on Systems, Man, and Cybernetics (SMC), pp. 1716–1721. IEEE, 2017a.
> >
> >  Yue Wen, Andrea Brandt, Jennie Si, and He Helen Huang. Automatically customizing a powered knee
> > prosthesis with human in the loop using adaptive dynamic programming. In 2017 International
> > Symposium on Wearable Robotics and Rehabilitation (WeRob), pp. 1–2. IEEE, 2017b.
> >
> >  Yue Wen, Jennie Si, Xiang Gao, Stephanie Huang, and He Helen Huang. A new powered lower limb
> > prosthesis control framework based on adaptive dynamic programming. IEEE Transactions on
> > Neural Networks and Learning Systems, 28(9):2215–2220, 2017c.
> >
> >  Yue Wen, Jennie Si, Andrea Brandt, Xiang Gao, and He Huang. Online reinforcement learning
> > control for the personalization of a robotic knee prosthesis. IEEE transactions on cybernetics,
> > 2019.
> >
> >  Ruofan Wu, Zhikai Yao, Jennie Si, and He Helen Huang. Robotic knee tracking control to mimic
> > the intact human knee profile based on actor-critic reinforcement learning. IEEE/CAA Journal of
> > Automatica Sinica, 9(1):19–30, 2021.
> >
> >  Dylan JA Brenneis, Adam S Parker, Michael Bradley Johanson, Andrew Butcher, Elnaz
> > Davoodi, Leslie Acker, Matthew M Botvinick, Joseph Modayil, Adam White, and Patrick M Pi-
> > larski. Assessing human interaction in virtual reality with continually learning prediction agents
> > based on reinforcement learning algorithms: A pilot study. arXiv preprint arXiv:2112.07774,
> > 2021.
> >
> >  Wu, Ruofan, Junmin Zhong, and Jennie Si. "Actor-Critic Reinforcement Learning with Phased Actor." arXiv preprint arXiv:2404.11834 (2024).
> >
> >  Michael Bain and Claude Sammut. A framework for behavioural cloning. In Machine Intelligence
> > 15, pp. 103–129, 1995.
> >
> >  Jennie Si and Yu-Tsung Wang. Online learning control by association and reinforcement. IEEE
> > Transactions on Neural networks, 12(2):264–276, 2001.
> >
> >  R. Hafner and M. Riedmiller, “Reinforcement learning in feedback control,” Machine learning, vol. 84, no. 1, pp. 137–169, 2011.
> >
> >  D. Silver, G. Lever, N. Heess, T. Degris, D. Wierstra, and M. Riedmiller, “Deterministic policy gradient algorithms,” in International conference on machine learning. Pmlr, 2014, pp. 387–395.
> >
> >  V. Mnih, K. Kavukcuoglu, D. Silver, A. Graves, I. Antonoglou, D. Wierstra, and M. Riedmiller, “Playing atari with deep reinforcement learning,” arXiv preprint arXiv:1312.5602, 2013.
> >
> >  Russell Enns and Jennie Si. Helicopter trimming and tracking control using direct neural dynamic
> > programming. IEEE Transactions on Neural networks, 14(4):929–939, 2003.
> >
> >  Sun, Jian, et al. "Direct heuristic dynamic programming based on an improved PID neural network." Journal of Control Theory and Applications 10.4 (2012): 497-503.
> >
> >  Yang, Qinmin, et al. "Reinforcement-learning-based tracking control of waste water treatment process under realistic system conditions and control performance requirements." IEEE Transactions on Systems, Man, and Cybernetics: Systems 52.8 (2021): 5284-5294.
> >
> > Gao, Ying, and Yan-Jun Liu. "Adaptive fuzzy optimal control using direct heuristic dynamic programming for chaotic discrete-time system." Journal of Vibration and Control 22.2 (2016): 595-603.
> >
> > El-Sousy, Fayez FM, et al. "Optimal Adaptive Ultra-Local Model-Free Control Based-Extended State Observer for PMSM Driven Single-Axis Servo Mechanism System." IEEE Transactions on Industry Applications (2024).

---

> > ### Comment · Reviewer_YG6k · 2024-11-26
> >
> > Thank you for the detailed comment.

---

### Author Response · Authors · 2024-11-25
**Global Response**

We greatly appreciate the Reviewers and the ACs for their time and feedback in reviewing our work.

We are very pleased that the Reviewers have recognized the significance of AIP for its several important contributions:

1. "The paper presents a clear, well-structured methodology with precise problem
formulation and experimental validation. The data-driven approach (AIP) is simple yet effective, demonstrating successful learning without simulation. Strong empirical results with clear metrics (20\% EMG reduction) and thorough ablation studies validating design
choices." (Reviewer tHbP)

2. "This can impact the wider adoption of machine learning techniques on such systems. A comprehensive analysis of the online adaption controller that shows significant improvement and personalized adaption." (Reviewer jBgV)

3. "The results presented are interesting and does justify the approach as a
practical solution to robot assisted locomotion." (Reviewer YG6k)

4. "Successfully getting online reinforcement learning to work on safety-critical
human-robot interaction task is commendable." (Reviewer YG6k)

5. "The paper addresses challenging problem of controlling soft exosuits for human assistance, ..., the method work directly in the real physical environment, without relying on a simulator. This makes the approach more practical and potentially more effective than sim-to-real methods." (Reviewer 1QdU)

6. "This paper .. without having a computer simulation to test things first - a significant challenge since most robotics work relies heavily on simulation before real-world testing. The paper presents a systematic investigation of the problem .. opening new possibilities for human-robot interaction in assistive devices. The researchers validate their approach with both theoretical analysis and practical experiments." (Reviewer tHbP)


For the issues raised by the Reviewers, we have taken great care to ensure all of the Reviewers' valuable feedback has been thoroughly addressed. A detailed explanation is provided in the individual responses below to each and every point raised by the Reviewers, and we will revise the manuscript accordingly.

>On Supplemental Video

First, we would like to emphasize that the results of the 4 participants included in the original submission were not cherry-picked. Namely, we used 4 participants in the study and we obtained 100\% learning success rate.

We have now included video clips of each of the 4 participants in supplementary materials.

As REVIEWER 1QDU requested data from more participants, we are making our best effort to collect a new dataset on a new participant, with which it brings our total number of participants to 5, comparable or more than the typical size used in most recent and relevant studies. The new data and related results that we have managed to process are included in Fig. 9 and Table 8 in the Appendix of manuscript. Please also note that, again, we obtained 100\% learning success rate for all 5 participants involved in the study, and the new data is consistent with previous results of the 4 participants.

>On Difference between Exoskeletons and Exosuits

We notice on occasions that the terms Exoskeleton and Exosuits were used interchangeably in the reviews. Although they both refer to wearable lower limb  technologies in the context of this paper, and are designed to augment human locomotion, but they differ significantly in several ways, from their structure, fabrication, to sensing, especially actuation. Exoskeletons are motor-driven rigid devices that provide physical support (directly via motor torque) and power to enhance strength, endurance, or mobility. They are often made of materials like metal or hard plastic, resembling a mechanical frame that partially or fully encases the body or limbs [Shi, Di, et al. 2019]. In contrast, exosuits are soft, lightweight, and flexible systems that use textiles, cables, and air inflation/deflation mechanisms for actuation to provide assistive movement without rigid structures. In terms of control, almost all exoskeletons are motor-driven. As the exoskeletons as ridig mechanical systems can be modeled through Euler-Lagrange mechanics, that makes model-based control methods (including classical control approaches) applicable [Sun, Yuanxi, et al. 2022]. However as we discussed in the related work, such model-based approaches are ENTIRELY not feasible for soft exosuit. Please refer to our Related Work section, lines 812 to 819 for more information. Controlling exosuits faces  unique new challenges. That is perhaps of the reason that soft exosuit control results are few in the literature even though conceptually one may prefer soft wearable such as exosuits over rigid mechanical devices such as exoskeletons.

---

> ### Author Response · Authors · 2024-11-25
> **Global Response cont**
>
> > On Diversity of Participants
>
> REVIEWER 1QDU suggested the study to involve "a larger and more diverse population, including people with different walking patterns, or physical impairments."
>
> First, assistive walking for unimpaired and impaired populations has been treated separately for good reasons. Assistive walking of unimpaired individuals (as what we have addressed in this study) aims at reduced effort yet achieving normative walking (just as that reflected in our paper title). Wearable robots for impaired patients, however, may have very different objectives. For incessance, typical for impaired populations, reduced effort is not of immediate concern. But rather, correcting gait pattern to improve gait symmetry, balance, and sometimes walking speed, among others, are of high priority. This is also partly due to strong data that irregular gait patterns of impaired individuals such as amputees can lead to secondary injury if their conditions are not corrected. Specifically, asymmetrical gait is frequently reported in people with unilateral lower limb amputation [Hof et al. 2007, Adamczyk et al. 2014], and is associated with many secondary issues, such as osteoarthritis of unamputated joints [Gailey et al. 2008] and lower back pain [Ehde et al. 2001]. With this said, at the high level, we still believe that our current AIP framework can potentially be extended to a new study to focus on assistive walking of physically impaired individuals. However, we will set up control design objectives differently (for sure not the same as the current objective), design offline data collection differently,  and we need a new experimental protocol to carry out the study, to name some among the tasks that we need to carry out. To be more specific, for example, Siviy, et al focus on poststroke patients while other studies [Li, Qinjian, et al. 2022, Li, Zhijun, et al. 2022, S. Sridar, et al.2020] only focus on unimpaired population for normative walking.

---

> ### Author Response · Authors · 2024-11-25
> **On Diversity of Participants cont**
>
> In terms of population size and diversity, and different walking patterns, we created the following table to summarize latest related studies. In the meantime, we provide a summary table of the participants involved in our current study.
> This comparison shows that 1) we used more  participants (total of 5) than other studies (total of 2, 3, 3), respectively in [Li, Qinjian, et al. 2022, Li, Zhijun, et al. 2022, S. Sridar, et al.2020]. 2) The participants in our study perhaps is the most diverse among comparable reports [Li, Qinjian, et al. 2022, Li, Zhijun, et al. 2022, S. Sridar, et al.2020]. While some more desirable details are not shown in [Li, Qinjian, et al. 2022, Li, Zhijun, et al. 2022, S. Sridar, et al.2020], but we have included men and women, tall and short, with rather large weight variance. 3) As for walking pattern, our videos clearly show visible differences in terms longer/shorter stride lengths, higher/lower frequency, among others. For one example, Participant 4 exhibits a noticeably smaller step length and shorter step duration compared to Participant 1.
>
> | Number                                                                      | Walking Type       | Gender  | Age (year) | Height (cm)  | Weight (kg)    | Robot Type    |
> |-----------------------------------------------------------------------------|--------------------|---------|------------|--------------|----------------|---------------|
> | C. Siviy et al. (IEEE Robotics and Automation Letters)  pariticipants data  | -                  | -       | -          | -            | -              | -             |
> | 1                                                                           | Poststroke         | N/A     | 33         | N/A          | 57.3           | Ankle exosuit |
> | 2                                                                           | Poststroke         | N/A     | 34         | N/A          | 91.5           | Ankle exosuit |
> | 3                                                                           | Poststroke         | N/A     | 54         | N/A          | 84.5           | Ankle exosuit |
> | 4                                                                           | Poststroke         | N/A     | 77         | N/A          | 101.8          | Ankle exosuit |
> | 5                                                                           | Poststroke         | N/A     | 49         | N/A          | 43.9           | Ankle exosuit |
> | 6                                                                           | Poststroke         | N/A     | 62         | N/A          | 66.3           | Ankle exosuit |
> | Li, Qinjian, et al (IEEE Transactions on Fuzzy Systems) participants data   | -                  | -       | -          | -            | -              | -             |
> | 1                                                                           | Normaltive Walking | Male    | 24         | 172          | 70             | Ankle exosuit |
> | 2                                                                           | Normaltive Walking | Male    | 25         | 175          | 65             | Ankle exosuit |
> | Li, Zhijun, et al. (IEEE Transactions on Robotics) participants data        | -                  | -       | -          | -            | -              | -             |
> | 1                                                                           | Normaltive Walking | N/A     | N/A        | 176          | 63             | Ankle exosuit |
> | 2                                                                           | Normaltive Walking | N/A     | N/A        | 175          | 70             | Ankle exosuit |
> | 3                                                                           | Normaltive Walking | N/A     | N/A        | 170          | 73             | Ankle exosuit |
> | S. Sridar, et al. (IEEE Robotics and Automation Letters) pariticipants data | -                  | -       | -          | -            | -              | -             |
> | 1                                                                           | Normaltive Walking | N/A     | N/A        | 172 +/- 5.35 | 76.33 +/- 0.94 | Knee exosuit  |
> | 2                                                                           | Normaltive Walking | N/A     | N/A        |              |                | Knee exosuit  |
> | 3                                                                           | Normaltive Walking | N/A     | N/A        |              |                | Knee exosuit  |
>
> Our paper data:
> | Subject | Gender  | Age (year) | Weight (kg) | Height (cm) |
> |---------|---------|------------|-------------|-------------|
> | 1       | M       | 26         | 76          | 175         |
> | 2       | F       | 27         | 52          | 154         |
> | 3       | M       | 28         | 79          | 165         |
> | 4       | F       | 31         | 57.5        | 158         |
> | 5       | M       | 28         | 80          | 172         |

---

> > ### Author Response · Authors · 2024-11-25
> > **On AIP's Contributions**
> >
> > >On AIP's Contributions:
> >
> > The CONTEXT. To provide a context of AIP's contribution to real time control application of offline to online RL (a topic that has been the focus of discussion with an extensive literature), representative/related methods and results include the following.
> >
> >
> > 1. Class 1, industrial robots.
> >
> > Typically operating independently in structured environments, these robots perform tasks such as grasping, assembly, and loading/unloading heavy objects in manufacturing settings. RL has been applied to improve adaptability and efficiency in these robots, enabling them to learn from human demonstration [Kathrin et al. 2010] and experience based on extensive simulations [Jens et al. 2013]. Recently RL has been used for motion planning [Meyes, Richard, et al. 2017], manipulation, and installation [Toner, Tyler, et al. 2023].
> >
> >
> >
> > 2. Class 2, human-robot interaction w/o robot physically attached to human.
> >
> > For instance, social robots have used RL to learn appropriate responses in interactive scenarios such as Maggie robot and icat robot [Neziha et al. 2021], to enhance user engagement [Sharif, Mohammadreza, et al. 2021]. In collaborative settings, RL helps robots learn from human feedback, optimizing their actions for better teamwork [Christiano, Paul F., et al. 2017]. These approaches allow robots to adapt in real-time, improving safety and efficiency in shared workspaces.
> >
> > 3. Class 3, wearable robots.
> >
> > This is a different problem from those in the previous two classes.
> > In this case, the robot is physically attached to the human user, thus the robot control is also subject to interactive dynamics between the human and the robot, a new aspect to be considered in robot control, that makes the problem complex and challenging. Furthermore, lower limb exoskeletons and soft exosuits differ significantly. While the exoskeletons can benefit from Euler-Lagrange mechanics to model its motion dynamics, and thus controlling exoskeletons is open to a broader possiblities of methods and approaches, from classical control to contemporary learning based control or a combination of the two. Specifically, the unique challenges of soft exosuit stem from 1) difficult or impossible to model the human-robot system, 2) difficult to quantify human-robot interacting dynamics, 3) a need of personalized control to meet individual users' conditions, to name some.
> > While in recent years, RL has advanced the control of lower limb wearable robots (such as robotic prosthesis and exoskeleton), these control problems are different from our current exosuit, namely soft wearable robot control. We have provided more details in lines 820 - 832, and in "Appendix B additional related work" section of our original submission.
> > Finally, it is worth mentioning RL control of lower limb prothesis [Yue Wen et, al. 2017c, Wu et al. 2021] that adapts assistance levels based on the user's gait patterns, as well as that of upper limb prosthetic where RL control has been deployed to refine control policies through user interaction, improving the intuitiveness and functionality of the exoskeleton devices [Ai, Qingsong, et al. 2021, Clautilde et al. 2020]. However, these applications either depend on simulator or readily accessible control torque of the prothesis or the exoskeleton. However in softsuit control under our consideration, neither a simulator nor a control torque is available for designing a control policy.
> >
> >
> > **In summary**, we thank the reviewers for the constructive questions that have given us the chance to highlight the true contribution of this work, which is NOT about re-inventing the behavioral cloning (BC) or direct heuristic dynamic programming (dHDP) methods (more details are provided in our response to REVIEWER YG6k). Those specific methods have existed and have stood the test of time. That was exactly the reason and what gave us the confidence to use these established methods DIRECTLY in real PHYSICAL environments involving human participants. As for the **contributions** of this work, we would like to emphasize that the proposed AIP method is the **FIRST** that directly online learns an optimal control policy from an offline policy in wearable soft robot. Our approach **WITHOUT** a simulator for extensive offline learning has few precedent in RL. It is also one of the few online RL approaches that has not only successfully demonstrated real time online optimal control but also has provided  performance guarantees in terms of learning convergence, solution optimality, and human-robot dynamic stability.
> > To the best of our knowledge, addressing several of these considerable challenges simultaneously is rare, not only in RL and robotics, but also more broadly in the controls literature.

---

> > > ### Author Response · Authors · 2024-11-25
> > > **List of Reference in Global Response**
> > >
> > > C. Siviy et al., "Offline Assistance Optimization of a Soft Exosuit for Augmenting Ankle Power of Stroke Survivors During Walking," in IEEE Robotics and Automation Letters, vol. 5, no. 2, pp. 828-835, April 2020, doi: 10.1109/LRA.2020.2965072.
> > > keywords: {Force;Legged locomotion;Optimization;Torque;Biology;Angular velocity;Textiles;Prosthetics and exoskeletons;rehabilitation robotics;wearable robots},
> > >
> > >
> > >  Li, Qinjian, et al. "Fuzzy-Based Optimization and Control of a Soft Exosuit for Compliant Robot–Human–Environment Interaction." IEEE Transactions on Fuzzy Systems 31.1 (2022): 241-253.
> > >
> > >  Li, Zhijun, et al. "Human-in-the-loop control of soft exosuits using impedance learning on different terrains." IEEE Transactions on Robotics 38.5 (2022): 2979-2993.
> > >
> > >  S. Sridar, et al. "Towards Untethered Soft Pneumatic Exosuits Using Low-Volume Inflatable Actuator Composites and a Portable Pneumatic Source," in IEEE Robotics and Automation Letters, vol. 5, no. 3, pp. 4062-4069, July 2020, doi: 10.1109/LRA.2020.2986744. keywords: {Actuators;Fabrics;Soft robotics;Finite element analysis;Strain;Soft robotics;soft actuator;wearable robotics;pneumatic source;soft exosuit}
> > >
> > >  Kober, Jens, J. Andrew Bagnell, and Jan Peters. "Reinforcement learning in robotics: A survey." The International Journal of Robotics Research 32.11 (2013): 1238-1274.
> > >
> > >  Meyes, Richard, et al. "Motion planning for industrial robots using reinforcement learning." Procedia CIRP 63 (2017): 107-112.
> > >
> > >  Toner, Tyler, et al. "Opportunities and challenges in applying reinforcement learning to robotic manipulation: An industrial case study." Manufacturing Letters 35 (2023): 1019-1030.
> > >
> > >  Sharif, Mohammadreza, et al. "End-to-end grasping policies for human-in-the-loop robots via deep reinforcement learning." 2021 IEEE International Conference on Robotics and Automation (ICRA). IEEE, 2021.
> > >
> > >  Christiano, Paul F., et al. "Deep reinforcement learning from human preferences." Advances in neural information processing systems 30 (2017).
> > >
> > >  Yue Wen, Jennie Si, Xiang Gao, Stephanie Huang, and He Helen Huang. A new powered lower limb
> > > prosthesis control framework based on adaptive dynamic programming. IEEE Transactions on
> > > Neural Networks and Learning Systems, 28(9):2215–2220, 2017c.
> > >
> > >  Ruofan Wu, Zhikai Yao, Jennie Si, and He Helen Huang. Robotic knee tracking control to mimic
> > > the intact human knee profile based on actor-critic reinforcement learning. IEEE/CAA Journal of
> > > Automatica Sinica, 9(1):19–30, 2021.
> > >
> > >  Ai, Qingsong, et al. "Machine learning in robot assisted upper limb rehabilitation: A focused review." IEEE Transactions on Cognitive and Developmental Systems (2021).
> > >
> > >  Nguiadem, Clautilde, Maxime Raison, and Sofiane Achiche. "Motion planning of upper-limb exoskeleton robots: a review." Applied Sciences 10.21 (2020): 7626.
> > >
> > >  At L Hof, Renske M van Bockel, Tanneke Schoppen, and Klaas Postema. Control of lateral
> > > balance in walking: experimental findings in normal subjects and above-knee amputees. Gait
> > > \& posture, 25(2):250–258, 2007.
> > >
> > >  Peter Gabriel Adamczyk and Arthur D Kuo. Mechanisms of gait asymmetry due to push-off
> > > deficiency in unilateral amputees. IEEE transactions on neural systems and rehabilitation
> > > engineering, 23(5):776–785, 2014
> > >
> > >  Robert Gailey, Kerry Allen, Julie Castles, Jennifer Kucharik, Mariah Roeder, et al. Review of
> > > secondary physical conditions associated with lower-limb amputation and long-term prosthesis
> > > use. Journal of rehabilitation research and development, 45(1):15, 2008.
> > >
> > >  Dawn M Ehde, Douglas G Smith, Joseph M Czerniecki, Kellye M Campbell, Dee M Malchow,
> > > and Lawrence R Robinson. Back pain as a secondary disability in persons with lower limb
> > > amputations. Archives of physical medicine and rehabilitation, 82(6):731–734, 2001.
> > >
> > >  Gräve, Kathrin, Jörg Stückler, and Sven Behnke. "Learning motion skills from expert demonstrations and own experience using gaussian process regression." ISR 2010 (41st International Symposium on Robotics) and ROBOTIK 2010 (6th German Conference on Robotics). VDE, 2010.
> > >
> > >  Akalin, Neziha, and Amy Loutfi. "Reinforcement learning approaches in social robotics." Sensors 21.4 (2021): 1292.
> > >
> > >
> > >  Shi, Di, et al. "A review on lower limb rehabilitation exoskeleton robots." Chinese Journal of Mechanical Engineering 32.1 (2019): 1-11.
> > >
> > >  Sun, Yuanxi, et al. "From sensing to control of lower limb exoskeleton: A systematic review." Annual Reviews in Control 53 (2022): 83-96.

---

### Note · Authors · 2025-01-24

I have read and agree with the venue's withdrawal policy on behalf of myself and my co-authors.